# How social networks affect the repression-dissent puzzle

**Shane Steinert-Threlkeld**[1‡], **Zachary Steinert-Threlkeld**[2‡*]

**1** Department of Linguistics, University of Washington, Seattle, WA, United States of America, **2** Department of Public Policy, University of California—Los Angeles, Los Angeles, CA, United States of America

‡ Authors listed alphabetically.
* zst@luskin.ucla.edu

**Data Availability Statement:** All of the data (both empirical and simulations) used in this paper are available on the Open Science Framework (http://dx.doi.org/10.17605/OSF.IO/6U9RW). At that project, there is a link to a GitHub repository that

## Abstract

Scholars have offered multiple theoretical resolutions to explain inconsistent findings about the relationship of state repression and protests, but this repression-dissent puzzle remains unsolved. We simulate the spread of protest on social networks to suggest that the repression-dissent puzzle arises from the nature of statistical sampling. Even though the paper's simulations construct repression so it can only decrease protest size, the strength of repression sometimes correlates with a decrease, increase, or no change in protest size, regardless of the type of network or sample size chosen. Moreover, the results are most contradictory when the repression rate most closely matches that observed in real-world data. These results offer a new framework for understanding state and protester behavior and suggest the importance of collecting network data when studying protests.

## Introduction

That state repression sometimes correlates with more protest but sometimes with less remains one of the most persistent puzzles in the subnational conflict literature. While the consistency with which states respond to threats is law-like [1], the responses to that repression vary across studies. Though severe enough repression should decrease protests [2, 3], it only sometimes does [4, 5]. Often, it does not [6, 7]. These inconsistent findings are especially puzzling because repression increases the cost of protesting, which should have a monotonically negative effect on protest size [2, 8]. Despite decades of "both better data and better reasoning" [9], the effect of repression on dissent remains puzzling.

These inconsistent findings—the "repression-dissent" puzzle—may owe to how regression models are constructed. Because social networks are a key determinant in individuals' decision to protest, we model protest and repression on social networks ("networks"), sample both the entire parameter space and empirically driven subsets of that space, and show that regressions consistently produce contradictory inferences. Three types of networks are analyzed, and the repression-dissent puzzle always emerges.

We use networks to model protests because individuals decide to protest based on the participation of their connections [10]. It is common to model individuals' decision to protest as a weighting of the costs and benefits of participating or not participating [2]. Individuals

contains the code for running the simulations reported in this paper, as well as scripts for analyzing the data.

**Funding:** The author(s) received no specific funding for this work.

**Competing interests:** The authors have declared that no competing interests exist.

estimate costs and benefits by observing how many of their connections protest, as the riskiness of protesting means that few individuals protest without first knowing others who are [11, 12]. Individuals protest if the number of protesting connections surpasses an internal threshold [13]. Surveys of protest participants confirm the importance of networks: a key motivating source is friends' participation, whether in the United States [14], German Democratic Republic [15], or Egypt [16]. Modeling protests using social networks therefore best captures the data generating process of them.

As the focus of this paper is on the effect of sampling, we use networks similar to those of [13, 17, 18]. Watts-Strogatz [19], Barabási-Albert [20], or Holme-Kim networks [21] are initialized, the state represses, mobilization occurs again, the state represses again, and so on until no new bystanders (inactive nodes) mobilize (become active). The severity of repression (the percent of protesting nodes removed during each repression step), the initial and final size of protests, and various network characteristics are recorded. We regress the final protest size on the repression rate and several control variables, using samples of the population of model trials. This process replicates how scholars collect data on repression and dissent: newspapers, the primary source of data for these studies [22], record a small subset of protest events [23].

Sampling the network simulation results recreates the repression-dissent puzzle. Regardless of the size of the sample or network type, the coefficient on the repression rate is sometimes statistically significant and positive or negative; other times, it is indistinguishable from zero. The repression-dissent puzzle emerges when sampling the entire parameter space of each network as well as subsets equivalent to values observed in actual social networks [24–26]. Using a large dataset of protest size, fatalities, and injuries, we show that the repression-dissent puzzle emerges most commonly in samples containing a realistic distribution of the rate of repression [27].

The emergence of the repression-dissent puzzle from these simulations is especially surprising, for two reasons. First, the simulations do not account for individuals' backlash to repression [18, 28]. In our model, repression can only decrease protest, yet regression based on the simulation results produces inconsistent inferences. Modeling backlash would therefore make the simulations' repression-dissent puzzle more pervasive in our results. Second, the main results include repression rates much higher than those observed in actual protests. The results are most contradictory—the percentage of regressions finding positive effects is closest to the percentage finding negative ones—at realistic levels of repression.

If modeling protest using networks is theoretically justified and recreates the repression-dissent puzzle when analyzed using current approaches, then researchers should reconsider those approaches. The most promising approach is to focus on measuring social networks before and during protests. New work has started to incorporate large scale network analysis into the study of protest: [12] uses cross-national network data to show that protest information has differential effects depending on the degree centrality of its source; [29] uses data on 130 million ties to show that "individuals are influenced by one another in social networks when deciding whether to participate in protests"; and [30] uses school attendance records to show how local network participation affects individuals' decision to protest. New work also shows how social network data can explain puzzling behavior in the study of intrastate conflict [31–33]. While survey work has often asked about protesters' ties to other protesters [15], only recently have these surveys been designed to elicit more complex network details [34].

## The persistence of the repression-dissent puzzle

Table 1 shows articles from the repression-dissent literature, their finding for repression, geographic and time resolution, and the size of the sample data. For repression, "-" and "+" mean

**Table 1. Mixed findings for repression's effect, regardless of sample size.**

| Study | Repression | Unit of Analysis | Observations (*n*) |
|---|---|---|---|
| [35] | +, - | Country-year | 50 |
| [4] | -,+ | Individual-year | 121 |
| [36] | +,- | Event | 2366 |
| [37] | +, - | Country-week | 194 |
| [38] | + | Country-week | 573 |
| [6] | -, + | Country-week | 52 |
| [39] | - | Event | 26-146 |
| [40] | +, - | Country-day | 2 cases |
| [28] | + | Event | 31 |
| [41] | - | Country-day | Hundreds |
| [42] | +,0 | Country-day | 540 |
| [7] | 0 | City-day | 700,435 |
| [12] | - | Country-day | 6,816 |

Note: + means backlash only.—negative. 0 means no effect. +,- n-shaped; -,+ means u-shaped. [4] consists of two surveys administered to the same individuals five years apart. Studies are arranged chronologically. None incorporate network measures.

statistically significant findings in those directions, while 0 means no statistically significant result. (This review is illustrative, not exhaustive; for a more thorough description of each, see Section S1 in S1 File) Note as well that the size of samples have increased over time, a feature of the puzzle which our results replicate.

Recent datasets that specifically code protest episodes and states' behaviors across dozens of countries also recreate the repression-dissent puzzle, as Figs 1 and 2 show. Fig 1 uses the Mass Mobilization (MM) dataset, which codes 13,060 protests with at least 50 attendees from 162 countries from 1990 through 2014 [43]. From that dataset, we model the size of protest as a function of protester violence and whether the state responds with accommodation, killing, arresting, shooting, or beating protesters; the model includes country and year fixed effects. (Protester violence and state response certainly interact with each other. While including them as independent regressors misses that dynamic, specifying and testing a more complicated model is outside this paper's scope. For a deeper analysis of those dynamics, see [44, 45], which use the MM data.) We then randomly sample half the observations, run a regression, store the *t*-statistics, and repeat that process 1,000 times. For both datasets, we drop observations that did not report protest size. These percentages are 32.57 and 34.66, respectively.

Fig 1 shows the results for arrests, killing, and shooting. Across regressions, the estimates for the effects are unstable. For example, in almost two-thirds of regressions, arrests negatively correlate with the number of protesters; most of the time, killing protesters does not have a statistically significant correlation, but it sometimes is associated with an increase or decrease in the size of protests. Shootings have similar inconsistent correlations. These inconsistent correlations are not true for all variables: in 994 regressions, protester violence negatively correlates with protest size (which matches others' findings [46–48]), and state accommodation is similarly consistent [6].

Fig 2 uses the Mass Mobilization in Autocracies Database v2.0 (MMAD), a dataset of 14,161 protests across 70 countries from 2003 through 2015 [49]. That dataset records the same inconsistencies. We model the size of a protest as a function of whether it is national or local, protester violence, and the level of state repression, in addition to country and year fixed effects. Fig 2 shows the results using the same repeated sampling process. The correlation of

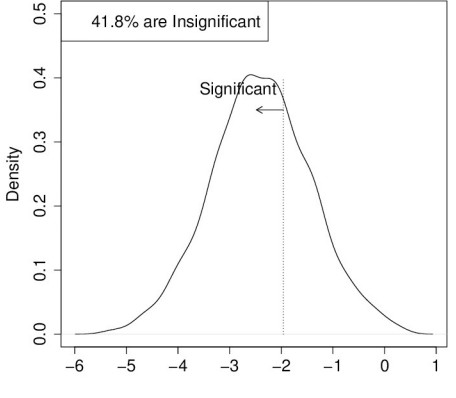

(a) Correlation with Arrest

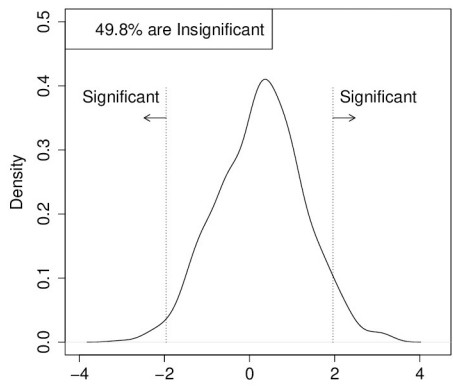

(b) Correlation with Killing

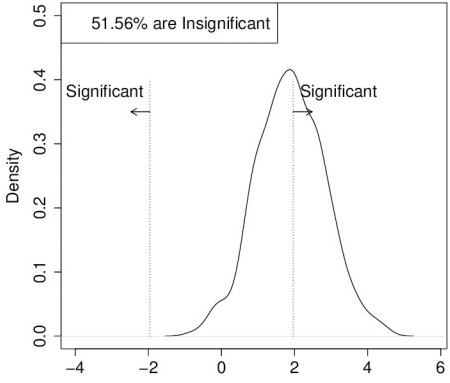

(c) Correlation with Shooting

**Fig 1. The repression-dissent puzzle persists in the mass mobilization dataset.** This figure is made using the Mass Mobilization (MM) dataset [43]. We run 1,000 ordinary least squares regressions that model the logarithm of the number of participants at a protest as a function of protester violence, state response (accommodate, arrest, beat, kill, or shoot protesters), and year and country fixed effects. Each sample contains 50% of the MM data. The t-statistic is recorded, and the histogram shows the distribution of the 1,000 values for arrests, killing, and shooting. Vertical dotted lines are at ± 1.96. These inconsistent findings *do not* apply to all variables: protester violence consistently correlates with smaller protests, state accommodation with larger, matching others' results [6, 46–48].

the three variables with protest size is inconsistent across samples; in some samples, a variable has a positive effect while in others it is negative.

The repression-dissent puzzle also persists when controlling for protester demands and the outcome is protest duration, as Figs A1 and A2 in S1 File show. (We considered modeling protester identity as well but did not. It is highly collinear with protester demands, and Mass Mobilization in Autocracies does not have identity variables. We also treat all protests occurring at the same time as distinct events.) The puzzle appears at various levels of temporal and geographic resolution and does not appear ameliorated when analyzed with newer, larger datasets. That the newest, largest datasets specifically constructed to measure protest and repression dynamics reproduce the inconsistent results of previous studies suggests that the repression-dissent puzzle constitutes an enduring feature of the literature on protest dynamics.

Neither the papers in Table 1 nor the datasets for Figs 1 and 2 incorporate network features. As we discuss more later, the inclusion of network data represents a promising avenue of future research.

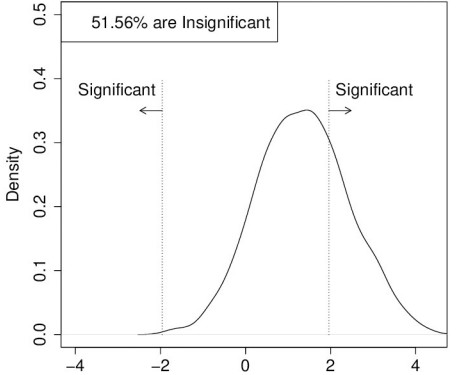

(a) Correlation with Participant Violence

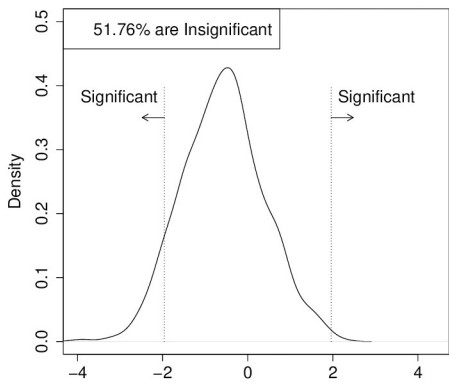

(b) Correlation with Scope of Demands

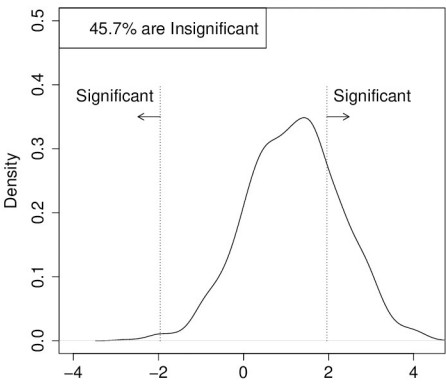

(c) Correlation with Repression

**Fig 2. The repression-dissent puzzle persists in the mass mobilization in autocracies dataset.** This figure is made using the Mass Mobilization in Autocracies (MMAD) dataset [49]. We run 1,000 ordinary least squares regressions that model the logarithm of the number of participants at a protest as a function of the scope of protester demands, protester violence, and state violence (no presence, presence, lethal intervention), and year and country fixed effects. Each sample contains 50% of the MMAD data. The t-statistic is recorded, and the histogram shows the distribution of the 1,000 values for participant violence, scope demands, security force engagement. Vertical dotted lines are at ± 1.96.

## Our model

There are three key features of human social networks: local clustering, short average path lengths, and highly skewed degree distribution. No extant model replicates this configuration, however, so we choose three—Watts-Strogatz, Barabási-Albert, and Holme-Kim—that capture combinations of the features. The Watts-Strogatz model replicates local clustering and short

**Table 2. Simulation parameters.**

| Parameter | Barabási-Albert | Watts-Strogatz | Holme-Kim | Notes |
|---|---|---|---|---|
| Individuals | 1,000 | 1,000 | 1,000 | Number of nodes |
| Threshold | $\sim \mathcal{U}(0,1)$ | $\sim \mathcal{U}(0,1)$ | $\sim \mathcal{U}(0,1)$ | Willingness to protest |
| Control Parameter | $\alpha = [2, 3]$ | $p = [0, 1]$ | $P_t = [0, 1]$ | Skew or clustering |
| Repression rate | $r = [0, 1]$ | $r = [0, 1]$ | $r = [0, 1]$ | Percent of protesters removed |

paths; Barabási-Albert short paths and skewed degree; and Holme-Kim local clustering and short paths as well as some skew. Table A1 in S1 File shows which models replicate which features, and Section S2 in S1 File explains their utility in more detail.

Table 2 summarizes the simulations' parameters. Only two parameters, *network control* and *repression rate*, vary across trials. For the Watts-Strogatz, *network control* refers to *p*, the percent of ties to rewire; for Holme-Kim, $P_t$, the probability of a triad formation per new edge. Each controls the amount of clustering in the network. For the Barabási-Albert network, the *network control* parameter is *α*, the slope of the power-law defining the distribution of degree centrality. As the *network control* parameter increases, skew and clustering (Watts-Strogatz) decrease while clustering in the Holme-Kim model increases. The *repression rate* is the percent of protesting nodes to remove during each repression step of the simulation. The *network control* in the Watts-Strogatz network varies from .001 to 1 in 20 log steps and from 0 to 1 in 21 linear steps for the Holme-Kim model. (Following [21], we set *m* to 3. $m_t$, the average of triad formation trials per step, then ranges from [0,1.8], which reproduces a wide range of clustering values.) It varies from 2-3, the range observed in human networks, in 41 linear increments in the Barabási-Albert model. The *repression rate* takes 101 values from 0 to .15 since repression almost uniformly stops protest above .15. Each combination of *network control* and *repression rate* is simulated with 1,000 trials.

All trials share the following three fixed features. The first feature is the distribution from which nodes' participation thresholds are drawn. These results use a uniform distribution on [0, 1], meaning some people may always protest and others may never protest; it represents the individual's willingness to join a protest and is inversely proportional to the amount by which an individual would benefit from policy change. Each node is randomly assigned a threshold from the uniform distribution, and this threshold is fixed within each trial but will differ across the 1,000 trials of each model configuration. Second, each model configuration contains nodes in one of the three network types. Third, each network contains 1,000 nodes.

Each trial runs as follows. At *t* = 0, an individual (the seed) is chosen randomly to become activated, and every node to which it is connected also becomes activated. Treating the start of protest as random is common in models of protest [50, 51]; in the implications section (subsection "Protest is the Error Term"), we provide a theoretical justification for this decision; and other threshold models use this initialization procedure [52]. At *t* = 0, we measure the mean and median degree of the initial protesters, clustering of the initial protesters including their connections who do not protest, average threshold of first protesters, the number of protesters, clustering within the initial protesters' networks, the average threshold within the protesters' neighborhood, the density of the network, the the clustering of the entire network, each node's degree and eigenvector centrality, the number of communities in the network, the number of communities with protesters, and the average percent of each community initially protesting. Communities are identified using the Louvain method of modularity maximization [53].

The state then represses by removing *r*% of protesting nodes, with each node weighted in proportion to its degree among all protesters. This weighting mimics states' targeted repression, the most common method of repressing protests [54, 55]. Two later robustness checks modify repression to weight by node threshold and degree so that the state prioritizes popular early adopters and to remove edges instead of nodes [56].

At subsequent time steps, protesters and individuals connected to them are evaluated. If the percentage of individuals to whom the evaluated is connected is greater than or equal to its threshold, it becomes activated. If the percentage is less than its threshold, it becomes deactivated. The state represses, ending each step. The simulation stops once no more nodes can be activated, and the total number of activated nodes is recorded as the final protest size. We also record how many steps each simulation takes before reaching this state.

This model is intentionally simple, for three reasons. First, the purpose of any model is to isolate the mechanisms of interest; our focus is the effect of repression, so we exclude complicating factors like security force loyalty [57], youth bulges [58], or the media environment [59]. Second, we focus on only two parameters, network control and repression, so that we can concisely explore the entire parameter space [60]. Because the simplicity allows us to inspect the entire parameter space, we consider it a virtue. Third, we draw from only a uniform distribution of thresholds because the distribution of thresholds otherwise introduces additional parameters to consider. We could find only one paper that attempts to measure thresholds, and it finds an approximately uniform threshold for online protests in Spain [61].

The simulations intentionally do not model backlash. Backlash, when repression causes more individuals to mobilize than it removes, is a theoretical explanation for the sometimes positive correlation of repression and subsequent protest size [28]. We do not incorporate backlash in order to emphasize the statistical foundation of the repression-dissent puzzle. Since backlash lowers the effective repression rate, including it would increase the number of larger protests, in turn increasing the range of $r$ over which the model reproduces the repression-dissent puzzle. Any result this paper finds for the repression-dissent puzzle is therefore a conservative estimate of the strength of the extent to which research design explains the repression-dissent puzzle. While backlash certainly explains some part of the repression-dissent puzzle, the puzzle's persistence without it suggests that other forces are at play.

## Simulation results

We now turn to presenting the results. First, we describe the analysis method: repeated regressions of random samples using theoretically motivated variables. After presenting the primary results, we show how the puzzle changes as a function of sample size and repression rate. The puzzle is especially prevalent when sampling with realistic repression rates and network parameters. The results hold when repression targets low-threshold popular nodes or is reconfigured to remove edges. Robustness checks show that inferences are consistent for many of the independent variables, meaning the puzzle does not exist due to a poorly specified regression. Clustering is discussed in more detail, and we confirm the puzzle holds with different versions of Eq 1, our regression model.

### Analysis method

We model the final size of protest as a function of the repression rate and network measures. We take samples of varying size, model protest size as a function of several variables, and record the $t$-statistic for the repression rate. This modeling is suggestive of the type of network features future data collection should try to record and some researchers have started to use [29, 62].

Eq 1 is the main regression model we use to model protest size.

$$
\begin{aligned}
\log_{10}(ProtestSize_i) = \quad & \beta_0 + \beta_1 * Density_i + \beta_2 * Global\ Clustering_i + \\
& \beta_3 * Protester\ Mean\ Degree_i + \beta_4 * Protester\ Median\ Degree_i + \\
& \beta_5 * Neighborhood\ Clustering_i + \beta_6 * Protester\ Clustering_i + \quad (1) \\
& \beta_7 * Initial\ Protest\ Size_i + \beta_8 * Network\ Control_i + \\
& \beta_9 * Repression\ Rate_i + \epsilon_i
\end{aligned}
$$

*Density$_i$* refers to the percentage of all possible connections in a network that actually exist. *Global Clustering$_i$* is the average of all nodes' clustering coefficient; the clustering coefficient is the percent of a node's connections that are themselves connected. *Protester Mean Degree$_i$* measures the average number of connections of the first protesters; *Protester Median Degree$_i$*, the median. *Protester Clustering$_i$* is the average clustering coefficient of the first protesters, and *Neighborhood Clustering$_i$* is the same, except the clustering coefficient is calculated using only ties within the first protesters (ties to non-protesters are dropped). *Initial Protest Size$_i$* is the number of protesters at $t = 0$, which is the degree of the seed protester plus one. *Network Control$_i$* is the parameter which controls the skew of network degree (Barabási-Albert) or amount of local clustering (Watts-Strogatz, Holme-Kim). *Repression Rate$_i$*, the treatment of interest, is the percent of nodes the state removes at the end of each time step.

These variables were chosen to capture various network features that affect protest size. Since most non-protesters require having connections with existing protesters [10], it is necessary to measure the number of ties in a network. We do this with the density variable as well as the three clustering ones. Controlling for clustering allows us to separate the effect of more ties randomly distributed and more ties distributed within groups of individuals more likely to know each other. We then measure clustering at the individual, neighborhood, and global level to further understand this distribution of ties [29, 62]. To ensure that the puzzle is not driven by accidentally choosing an influential seed protest, we record the mean and median degree of the first protesters. Since clustering and influence in a network are determined by a tunable control parameter, we control for it. Finally, the initial size of protests is a reliable determinant of their final size [30, 43] and is partially a lagged dependent variable, so we include it as well.

For each regression, we vary the size of the sample from 50 to 2000 in increments of 50 and record the *t*-statistic on each variable; each sample size is repeated 1,000 times. Increasing the sample size replicates the increasing number of observations in modern studies, as documented in Table 1. This step ensures that any conclusions about the repression-dissent puzzle reflect true contradictions in the underlying process, not small sample-size limitations. For each sample size, we therefore have 1,000 *t*-statistics for the effect of repression. This process is applied separately to the Watts-Strogatz, Barabási-Albert, and Holme-Kim models. At no point do we use any data from the studies in Table 1: except to determine realistic repression rates, the rest of the paper only analyzes the simulation data.

Once the *t*-statistics are recorded, we measure the percent of regressions for which the regression coefficient is statistically significantly greater than 0, less than 0, or indistinguishable from zero, separately by network type. A statistically significant negative correlation corresponds to repression working, and a correlation indistinguishable from zero suggests that other factors, such as the initial size of protest or topographical network characteristics, determined the final protest size. A statistically significant and positive correlation means increasing repression is associated with more protest. (Since the model does not contain backlash, any positive correlation between repression rate and protest size will only exist in small samples. We do not keep the individual samples used for each regression and so cannot investigate these few occurrences, but we suspect they occur when high *r* occurs with larger enough initial protests that removal is not strong enough.) This *y*-axis is plotted against the sample size.

All regressions use ordinary least squares. Standard errors are not clustered because each simulation is independent and the network types are analyzed separately. The level of significance is.05 using a two-tail test. The Supplementary Materials reports results using negative binomial models; those results do not change.

## Primary results

Fig 3 presents the main results, with subfigures ordered from least to most supportive of the repression-dissent puzzle. Fig 3a replicates Figs 1 and 2 for sample sizes of 1000. Though this histogram only finds inconsistency for the Barabási-Albert model, it is the one least likely to recreate the puzzle because it does not show results as sample size changes or at realistic values of repression. Fig 3b shows that inferences change as a function of sample size, revealing that the repression-dissent puzzle exists on all three networks but erodes as the sample size increases.

The first two panels equally sample all values of *r*, biasing against the emergence of the repression-dissent puzzle because rates of repression greater than.025 occur in only 15% of

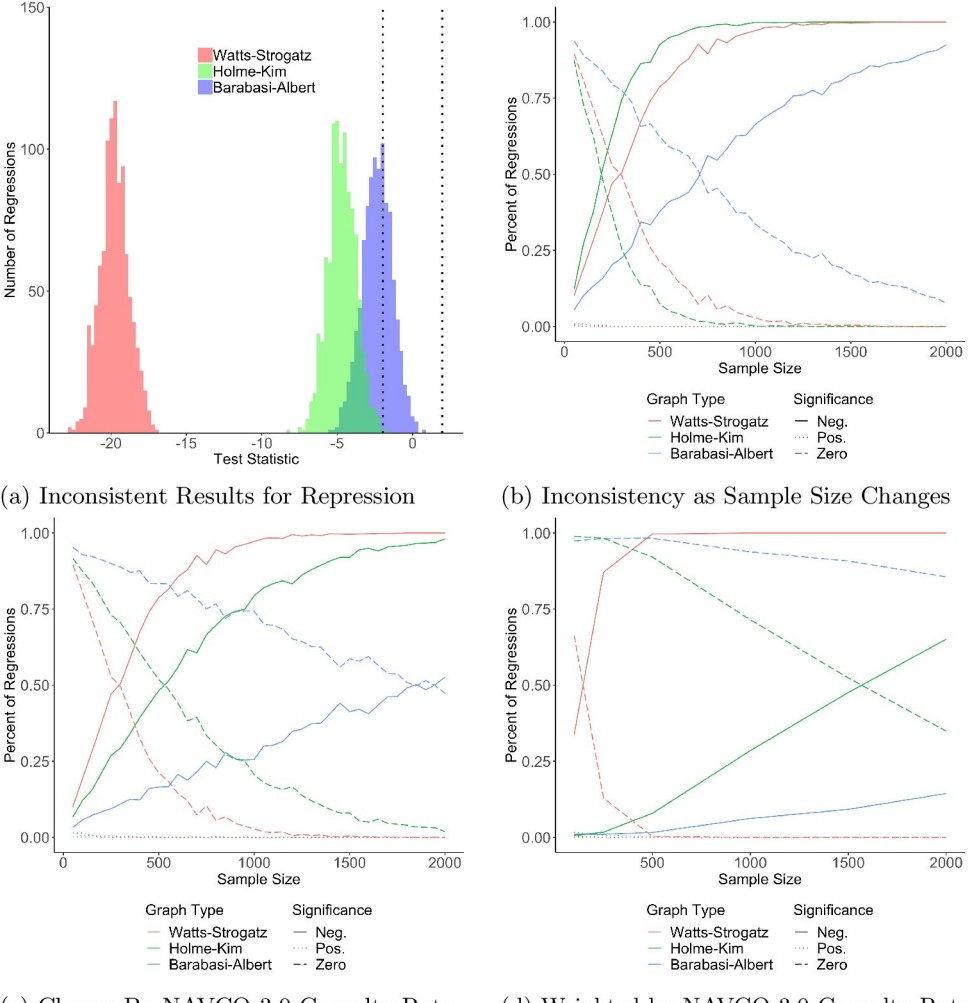

(a) Inconsistent Results for Repression

(b) Inconsistency as Sample Size Changes

(c) Chosen By NAVCO 3.0 Casualty Rate

(d) Weighted by NAVCO 3.0 Casualty Rate

**Fig 3. The repression-dissent puzzle as a function of sample size.** For the three network types in Table 2, 1,000 trials are run for each combination of repression rate and network control. For all, we use the regression model in Eq 1 and record the *t*-statistic of each coefficient. For (a), we take 1,000 samples of size 1,000 per network type and show the distribution of *Repression Rate$_i$*'s t-statistic for each network type. The vertical lines are at ±1.96, traditional thresholds for statistical significance. For (b), we vary the sample size from 50 to 2,000 in increments of 50 and record the percent of t-statistics that are positive, negative, or not significant. (c) repeats (b) but limits $0 \leq r \leq .1005$, the average casualty (fatality plus injury) rate observed in NAVCO 3.0. (d) samples the trials using repression weights generated from NAVCO 3.0. The more realistic samples in panels (c) and (d) evince the strongest evidence for the repression-dissent puzzle.

protests, based on an analysis of the Nonviolent and Violent Campaign Outcomes (NAVCO) 3.0 dataset [27]. Sampling regardless of the model's repression rate generates a sample where 83.33% of protests have a repression greater than or equal to.025, 33 times more than there should be.

Panels (c) and (d) show that increasing the sample size, however, is not the solution to the repression-dissent puzzle once realistic rates of repression are taken into account. Panel (c) shows the results restricting samples to $r \leq .1005$, the average casualty rate (fatalities plus injuries) NAVCO 3.0 records. Panel (d) draws samples based on the actual distribution of casualty rates, favoring low rates of repression. It therefore represents the distribution of repression rates in the simulation samples which most closely resembles the distribution of repression rates in actual data. In both sampling variations, the repression-dissent puzzle is stronger, i.e. the lines for negative and zero statistical significance cross at larger sample sizes than in (b). Panel (d), the most realistic sample of repression rates, especially suggests that the repression-dissent puzzle is only partially resolved with increasing sample sizes.

### Effect of repression rate and sample size

Fig 4 presents the repression-dissent puzzle as a function of the repression rate and three sample sizes. Here, we randomly sample $n$ trials where $Repression\ Rate_i \leq r$ and record the $t$-statistic for $\beta_9$. We repeat this process 100 times for 100 values of $r$ between $[0, .15]$. We also vary the size of the sample, $n \in \{1000, 1500, 2000\}$. For each combination of $n$ and $r$, we therefore have 100 $t$-statistics for the effect of repression. This process is applied separately to the three network models. The shaded gray region emphasizes the range of repression rates that are most realistic, based on NAVCO 3.0's data: the mean fatality rate is 5.4% (left edge), casualty 10.05% (right edge). Focusing on the gray regions reveals that the simulation is most likely to generate

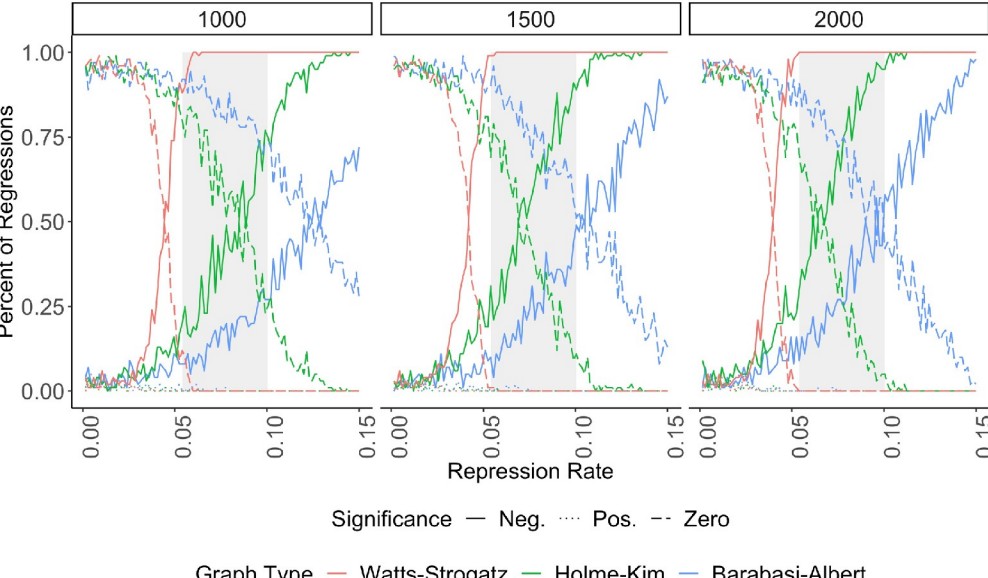

**Fig 4. The repression-dissent puzzle as a function of sample size and repression rate.** We generate trials and results in the same manner as Fig 3. Here, we sample $n$ trials (facet label) where $Repression\ Rate_i \leq r$ (x-axis). The $t$-statistic is recorded, and this process repeats 100 times for 100 uniformly spaced values of $[0 \leq r \leq .15]$. The y-axis is the percent of regressions which find no relationship or a statistically significant (positive or negative) one for $r$. The grey box emphasizes the range of realistic values for $Repression\ Rate_i$; its left edge is the average fatality rate and its right is the average casualty rate, measured using NAVCO 3.0.

contradictory results at realistic rates of repression. Presenting results as the maximum value of $r$ increases generates samples that more closely resemble real-world protest data, a point emphasized when discussing Fig 6.

The repression-dissent puzzle is consistent across network structures and sample sizes. Figs 3 and 4 show that repression always succeeds (has a statistically significant negative effect on the size of protest) most often on Watts-Strogatz models, followed by Holme-Kim. Repression succeeds least frequently on Barabási-Albert networks. Regardless of the network type, however, the repression-dissent puzzle exists across a range of sample sizes and repression intensities.

## Realistic network parameters and repression rates

Fig 5 replicates Figs 3 and 4 but with network configuration parameters most similar to real social networks. The Barabási-Albert network uses only trials where the network scaling parameter, $\alpha$, equals 2.3. This value was chosen based on analyzing the follower distribution from 100 hours of random Twitter data and is near the 2.276 documented in [63]. In addition to Barabási-Albert where $\alpha = 2.3$, we keep only Watts-Strogatz trials where $p = .34$ and Holme-Kim ones where $m_t = .315$. For these results, the simulations are subsetted to keep only trials where $\alpha = 2.3$, $p = .34$, or $m_t = .315$.

The values of $p$ and $m_t$ were chosen to produce a realistic global clustering coefficient. The most comprehensive measures of clustering we could find are from [24, 25], where the clustering of an e-mail network of 5,000 university students is reported as .16. Using non-electronic communication data, [26] finds a similar value for residents of Framingham, Massachusetts; [64] finds one of .07 for 1,000 Danish college students; and [65] finds one of .25 using call detail records for a 20% sample of Portugal. Since studies have not found a stable value for clustering, a value of .16 is a reasonable compromise.

Panel (a) shows the results as sample size changes, and (b) shows them as the repression rate changes. In both panels of Fig 5, the repression-dissent puzzle persists. When not restricting by repression rate (a), the Watts-Strogatz models nearly always find a negative effect for repression. The behavior of the other two networks is little changed when not restricting by network parameters. When restricting by repression rate (b), the repression-dissent puzzle exists for all sample sizes and network types, especially at realistic rates of repression (the grey rectangle).

Fig 6 shows the distribution of fatalities and casualties recorded in the NAVCO 3.0 dataset, the most comprehensive dataset that contains estimates of protest size and casualties. We first keep only protest events (verb_10 == 14). Of these 15,170 events, 9,095 contain a value for the num_partic_event variable. We discard the 2,446 of those events whose size is reported as a range, e.g. 70,000-200,000. Of all protest events, 26.89% contain missing values, but discarding those with a size reported as a range increases that number to 56.2%. (The high estimate excludes protests where injuries or fatalities are not recorded. The low estimate treats those missing data as true zeros.) The events where everyone is killed or injured are small: an average size of 1.3 for the former, 24.04 for the latter. Because NAVCO 3.0 relies on newspapers and newspapers select for violence, the measured repression rates are most likely greater than the true, but unknown, repression rate. The median and mode rates are 0. The distribution suggests that average fatality rates range from 1.06%-5.75%; casualty rates are 1.81%-10.05% [66]. The gray box in figures with $r$ on the $x$-axis emphasize the realistic ranges of repression.

The real-world distribution of repression rates means the simulations' results should not be interpreted as unambiguous effects for repression because only Fig 3d samples data using a

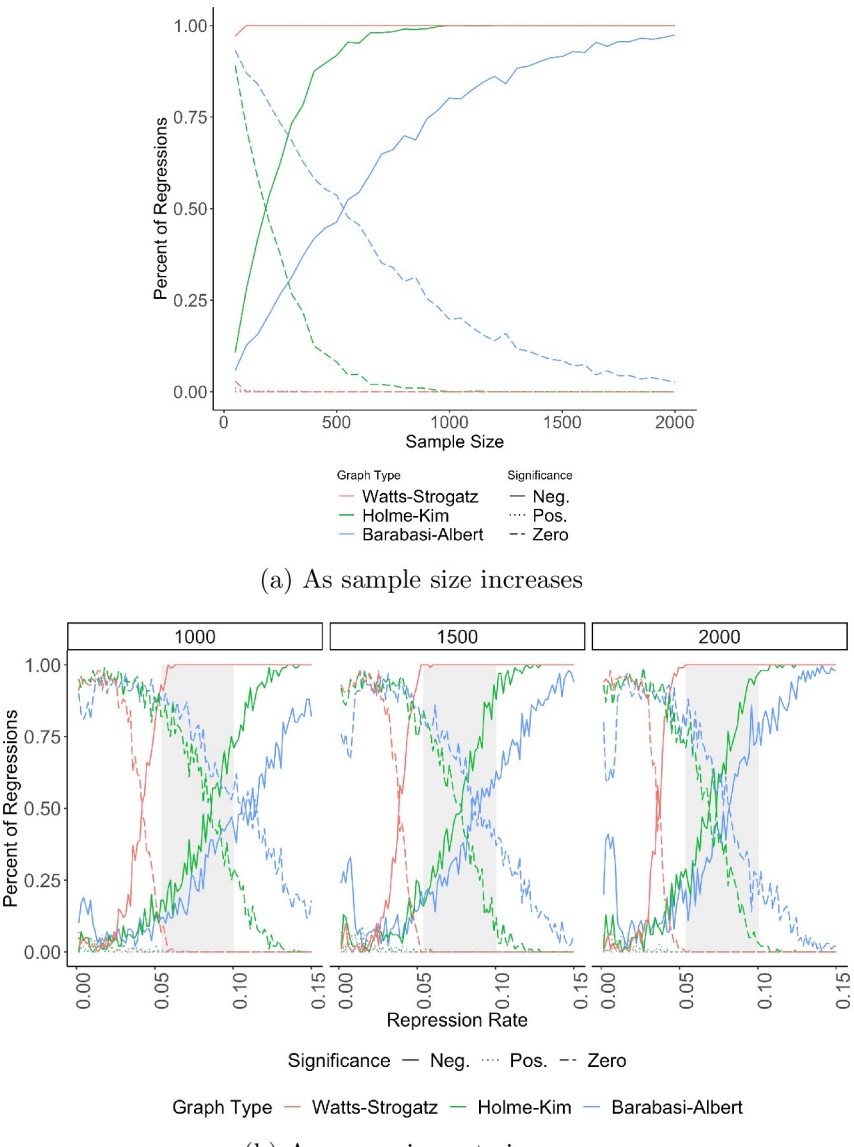

(a) As sample size increases

(b) As repression rate increases

**Fig 5. The repression-dissent puzzle exists at realistic network values.** For each network, the value of $p$, $P_t$, or $\alpha$ is chosen according to results reported in [24, 25, 63] (a) shows as the sample size increases; (b), as the repression rate increases, holding sample size constant at three different values. The grey box emphasizes the range of realistic values for *Repression Rate$_i$*; its left edge is the average fatality rate and its right is the average casualty rate, measured using NAVCO 3.0.

realistic distribution of *r*. The rest randomly sample the entire range of *r*. The model's results in those ranges of repression show regressions with statistically significant and not statistically significant results occur frequently; neither the positive nor negative effect dominate. At realistic rates of repression the simulations suggest the repression-dissent puzzle is strongest. Not modeling backlash and not allowing nodes to form edges during a protest [17] also biases against the repression-dissent puzzle emerging.

Once approximately 15% of protesters are repressed, repression almost always succeeds, suggesting sufficiently severe repression unambiguously decreases protest. Though 15%

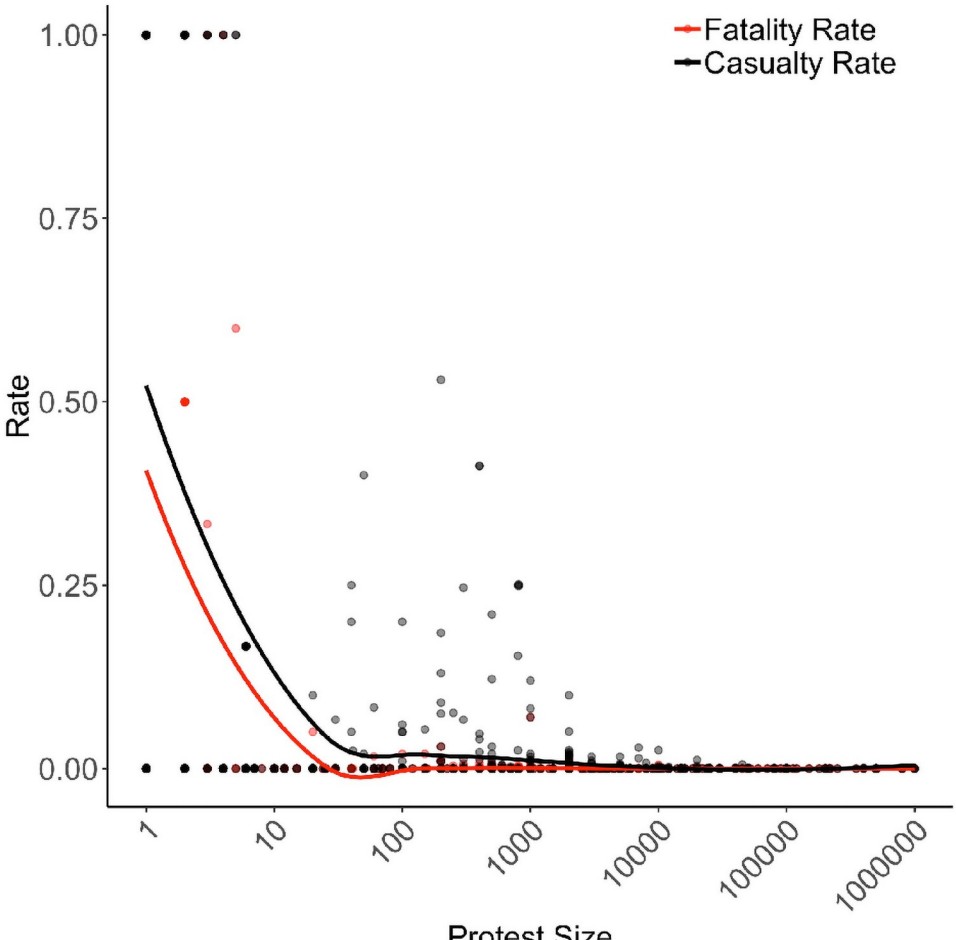

**Fig 6. Empirical distribution of protest casualties, 1991–2012.** The distribution of protest fatalities and casualties according to the Nonviolent and Violent Campaign Outcomes 3.0 dataset [66]. Represented are 6,649 protests from 05.10.1991 to 12.31.2012. Normalizing protest size makes both lines appear flat: the fatality and casualty rates are approximately zero for protests larger than 1,000 (.01), the first percentile of the size distribution.

sounds low, this rate is quite high. Only 5.76% of NAVCO 3.0's events have a fatality rate of at least 15%; including injuries, the percent increases to only 11.60. [28] catalogues 31 massacres of protesters during the 20th century; of those, the average fatality rate is 4.93%, and only 9.67% of events have a fatality rate of at least 5%. Massacre's average casualty rate is 12.22%, and only 35.48% of massacres have a casualty rate of at least 5%. Real-world repression rates are therefore much closer to 0 than 1. The repression-dissent puzzle that emerges from this model therefore is not due to focusing on unrealistic values of repression. Realistic values of repression are where the repression-dissent puzzle emerges most strongly.

## Alternative repression mechanisms

Finally, Fig 7 uses two other implementations of repression. First, we incorporate a node's threshold into the construction of its probability of repression so that low-threshold nodes are more likely to receive repression. We weigh the active nodes as follows. First, we compute the degree centrality of each node, normalized to [0, 1] by dividing by the largest value. Then, we normalize the nodes' thresholds to [0, 1] by dividing by the largest value. For each node x, we

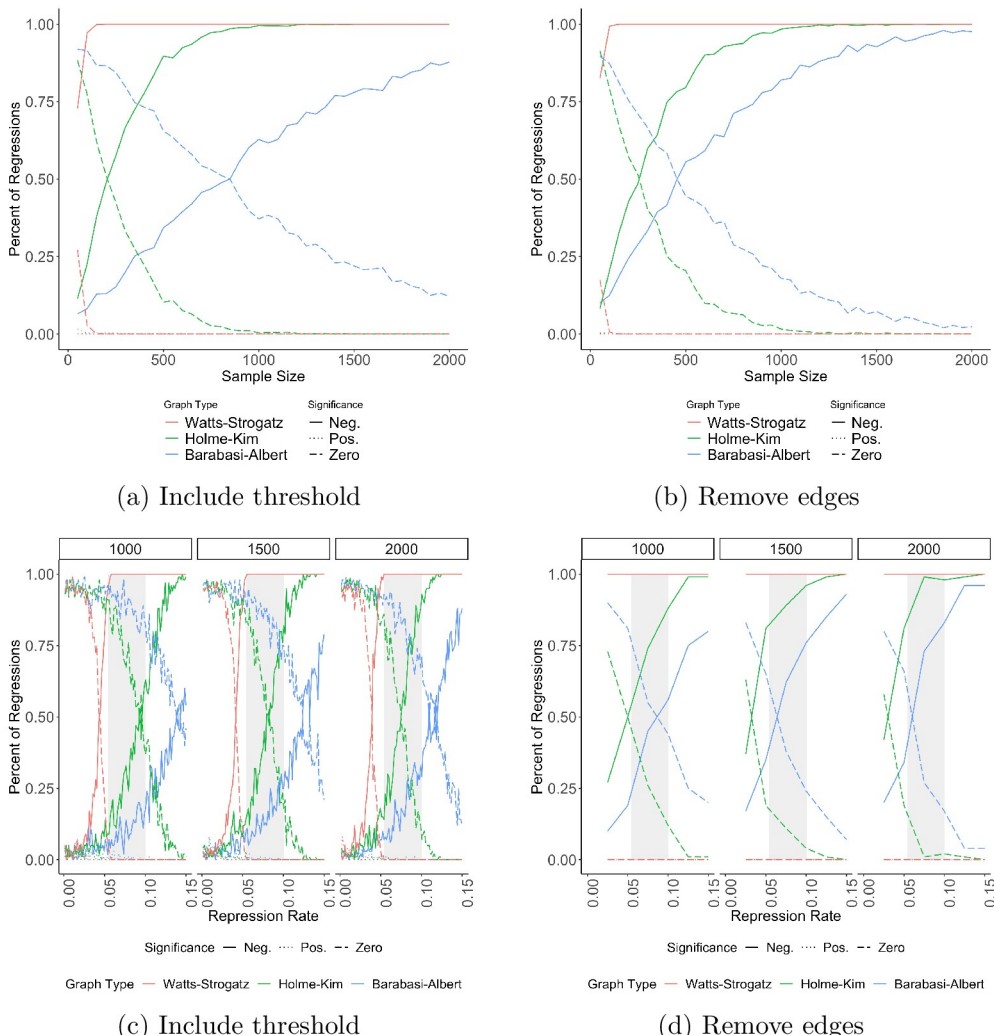

**Fig 7. The repression-dissent puzzle exists with different repression implementations.** (a) shows the result when weights used to select nodes to repress take into account the node's threshold, with lower threshold nodes more likely to suffer repression. (b) removes $r$ percent of a node's edges.

let its $w_x = d_x + (1 - t_x)$, where $d_x$ is the normalized degree and $t_x$ is the normalized threshold. This weight has the effect of balancing degree centrality with threshold: nodes with high thresholds are less likely to be repressed, while those with lower thresholds are more likely. Second, we model repression as edge removal [56]. In these simulations, $r$ corresponds to the percent of a node's edges removed, allowing a node to influence others even if it suffers repression. Both types of repression recreate the repression-dissent puzzle.

## Robustness checks

Section S4 in S1 File presents three further verification checks. The first two verify that social network features, not model construction, generate the repression-dissent puzzle. Fig A3 in S1 File shows that the results hold when restricting analysis to only protests which grow in size; this subsetting therefore ignores protest where repression may have failed because there were not enough targets. A second concern is that the method for creating initial protests drives the

contrasting regression results; this worry is especially pressing for Barabási-Albert networks since their skew is so pronounced that some protests will initialize with hundreds of participants, at which point repression will almost always fail. Fig A4 in S1 File shows that the repression-dissent puzzle holds when restricting trials to those where the initial protest size is less than or equal to the median protest size and protest size changes. (Without the growth requirement, over 90% of regressions find no correlation between repression and protest size.) The third verification, Fig A5 in S1 File, shows that the choice of estimator does not drive results; the findings are the same when using a negative binomial estimator on the untransformed data.

This paper's recreation of the repression-dissent puzzle is not because the data or regression generate ambiguity for all variables. Figs A6-A8 in S1 File show the distribution of $t$-statistics for the non-repression variables in Eq 1 and two threshold variables, the mean threshold of protesters and the mean threshold of their neighbors. Across network types, there are consistent results for the average threshold of initial protesters (−) and protest's initial size (+). Barabási-Albert and Holme-Kim networks have the same relationship for initial neighborhood clustering (+) and global density (−). All three networks find contradictory results for the degree of initial protesters, clustering within initial protesters, the average threshold of neighbors of initial protesters, and global clustering.

Section S4.1 in S1 File explores the effect of local clustering on the repression-dissent puzzle. Fig A9 in S1 File compares Barabási-Albert trials where $\alpha$ = 3 to Holme-Kim trials, which have $\alpha \approx 3$ but with tunable local clustering. Comparing the Holme-Kim and Barabási-Albert trials, the difference in repression rate at which the percent of regressions with negative or neutral results equals is even greater than in Fig 3. This behavior suggests that clustering makes repression more effective. Fig A10 in S1 File shows how inference is sensitive to changing clustering (Watts-Strogatz, Holme-Kim) or skew (Barabási-Albert) of a network. As Watts-Strogatz networks transition from regular to small-world networks ($0 < p \leq .1$), the probability that the rewiring rate affects successful protest is indeterminate. (Centola2007 and [67] find this non-monotonicity across the range of $p$. Our inference differs probably because we do not vary $p$ in small enough steps to observe a consistent effect as it increases across small values.) As local clustering approaches that of a random network ($p \to 1$), the probability of protest succeeding increases then decreases. Protests on Holme-Kim networks are not sensitive to increases in local clustering. The scaling parameter of Barabási-Albert networks exhibits a similar puzzle as the repression rate: as it increases (as the network becomes less skewed), protest becomes more likely to succeed.

We have also confirmed that the results are not sensitive to model specification. An additional model includes variables for the average threshold of initial protesters and the average threshold of their neighbors. Figs A6 and A7 in S1 File show those results. Higher thresholds for initial protesters negatively correlate with protest size, while the average threshold of protesters' neighbors does not. The repression-dissent puzzle also emerges from more parsimonious models. Whether the regression includes only a variable for repression, adds one for the protest's initial size, or excludes global network variables, the repression-dissent puzzle persists. Since results are almost identical, the Supplementary Materials do not show them, but they are available upon request.

## Implications for understanding repression and protest

The ability to replicate the repression-dissent puzzle has several theoretical and empirical implications for the study of protest.

## Repression and loyalty

This paper's results show that repression is most effective as networks become more clustered and have less degree skew. In a tyrant's dream, society would consist of small groups of individuals with no connections outside of their unit: families who know no other families. Regime strategies are therefore designed towards this "ideal" state of no connections across small groups.

Small-world networks, a special case of the Watts-Strogatz model, are the closest to this ideal because nodes maintain relatively few connections and few of those connections are not clustered together. Figs A9 and A10 in S1 File are particularly instructive here. The former shows that a Barabási-Albert model when the degree skew is the same as the Holme-Kim is much more likely to reveal the repression-dissent puzzle; the only difference between these two is that Holme-Kim has higher local clustering. The latter shows that, in the Watts-Strogatz model, more clustering makes repression more effective but only up to a point: too much rewiring removes local clustering, and repression becomes more indeterminate. Moreover, in all results the repression-dissent puzzle is the strongest in the Barabási-Albert models, which have the least clustering and most degree skew.

Many repressive policies can be better understood by analyzing their affect on clustering and degree skew. Monitoring personal communications or relying on informants are examples of policies that make it harder to trust individuals outside of very small friend and family units. As domestic spying increases, individuals then inhabit smaller and smaller social worlds, making it harder for protest to spread. Restricting independent civil society, the media, or mass gatherings are examples of policies that inhibit the formation of connections across communities and reduce degree skew, making it more difficult for protests to jump communities. Regimes often allow protests in cordoned, easily controlled physical spaces, such as journalists' syndicates [68] or university campuses [69]. Since those environments have few exits and include people from similar social groups, these protests are unlikely to grow.

Figs 3 and 4 show that repression's effects are nonlinear, regardless of network type. This result suggests that many types of repression may have no effect because they are not sufficiently severe, even if states and protesters perceive different tactics as increasingly severe. For example, a state may progress from using water cannons to arresting protesters to firing rubber bullets. If those tactics all remove a small percentage of protesters, even if they remove increasing percentages, they will not produce statistically observable reductions in protest size.

Many behaviors of repressive governments are designed not to alter network structure but to shift thresholds since loyalty to the status quo increases an individual's high participation threshold. Thus, even if a country's social network becomes less clustered and has more influential individuals than those in charge would like, high thresholds introduce friction to slow the spread of protest. Regime policies that favor one class, race, or ethnicity over another have the effect of pushing the benefited's thresholds closer to 1. Policies such as subsidies, state jobs, and propaganda have the same effect, decreasing the probability of large-scale mobilization. Requiring membership in the ruling party has the same effect, as does creating a large bureaucracy. Fig A6d in S1 File substantiates this strategy of increasing citizens' participation threshold.

The importance of such behavior can be seen in the cases of Saddam Hussein and Muammar Gaddafi. The former built an institutional apparatus that inspired some loyalty in the face of an invasion and allowed him to remain at large for months. The latter did not pursue policies to increase loyalty in large segments of the population, used mercenaries, lost power quickly, and was brutally killed. These behaviors are substantiated by formal models showing

that greater threshold variance makes it more difficult for a protest to start [13, 70, 71] as well as in Fig A6d in S1 File.

Moreover, the more a regime manipulates a society's social network, the more extreme any subsequent mobilization will be. Precisely when a regime makes it more difficult to observe others when making a decision, individuals can only act when they know a very large number of others have also seen whatever information they are also processing [72]. Thus as regimes induce loyalty in segments of the population and atomize society, the more likely are initial protests to be large: there can be no other kind. For example, in the German Democratic Republic, one-third of the population protested; in Czechoslovakia, 70% [37]. More recently, approximately 20% of Bahrainis protested during the Arab Spring [73]. Our model replicates these results, as Fig A7a in S1 File shows.

## Data bias

Though this paper's models reproduce inconsistent results found in the empirical literature, they generate many fewer episodes of repression correlating with more protest than the empirical literature observes. Part of this result is the lack of backlash. Some of the result is also due to how events become entries in event datasets.

If an event dataset relies on newspapers, it replicates well-known biases. (For thorough introductions to these biases, see [74, 75]). One of these biases is a preference for reporting on violence ("if it bleeds, it leads"). For repression to be large enough to be observed by newspaper reporters, protest must reach some threshold size. Most of this model's cases of successful repression, however, are on protests that start small and never grow in size. These "protests" are equivalent to successful preventative repression or repression of small protests, which are recorded much less rarely in newspapers. That existing datasets appear to record repression decreasing protest size less frequently than the model may partly be due to the fact that most of the successful episodes of repression do not appear in newspapers. New work using Sina Weibo and Twitter to measure protests finds that traditional media significantly underreport the true number of protest events [48, 76].

## Protest is in the error term

Just as war is in the error term [77], individual occurrences of protest may be fundamentally unpredictable. Scholars have continuously refined their understanding of the characteristics making states more or less likely to start a war, but randomness continues to dominate models predicting war outbreak [78]. Similarly, empirical advances have given scholars of subnational conflict better understanding of which variables correlate with the start of protests, civil wars, and coups [79–82]. If states engage in repression of potential protests, then any that occur appear randomly [7]. If protest is in the error term, then repression's success may be in the error term as well. States may randomly fail to prevent a protest, and once a protest starts, it can grow more quickly than repression shrinks it.

Activists implicitly realize this dynamic. For them, if the difference between a benign protest and a revolutionary one is unknowable, the prudent course of action is to protest as often as possible. This randomness explains why Leipzig activists organized weekly meetings for years, why Asmaa Mahfouz was not deterred by her failed protest on January 18, 2011 in Cairo, and the importance of continuing to protest throughout the Deep South throughout the Civil Rights movement.

If mass protests reflect an accumulation of grievances over time, it is easy to see why *post hoc* analyses of protest, and conflict more broadly, find missed signals [50]. For a state's actions to generate the potential of protest mobilization, it has to undertake many behaviors which

analysts can later call causes. For example, protests in East Germany were motivated by some mixture of environmental politics and economic grievances, and the Berlin Wall was opened due to bureaucratic miscommunication [83]. In Tunisia and Egypt, predatory police, hereditary politics, economic stagnation, and rising living costs comingled, though in Egypt elite defection may have been the pivotal factor [84]. Specific large mobilizations will never be predictable; if they were, they would not occur.

The simulated results concur with qualitative descriptions of mass protests. It is not uncommon for major protests to start from small events, events which in previous times in the same place faded away. For example, Detroit's 1967 riots started in response to a raid on an illegal bar, and Tunisia's Arab Spring protests, which inspired subsequent protests throughout the Arab world, started in response to a provincial fruit vendor's immolation after police seized his wares. The modern Tea Party movement in the United States started after a rich broadcaster (Richard Santelli) yelled at rich traders on the Chicago Mercantile Exchange. "It was just a moment", he explained, five years later [85]. A network explanation helps understand how small actions can scale to large protests, making those protests seem surprising because they have innocuous, random beginnings.

### Structural variables

Grounding explanations in network structure has implications for how scholars approach structural explanations of protests. For example, it is possible that demographic factors, especially large populations of youth, make protest onset more likely [58]. The situation may become more volatile if those youth are unemployed, especially if they are educated [86]. If each young, educated, and unemployed person lives alone and cannot socialize, however, it is difficult to envision how those structural factors would cause protest. That structural effects are inputs to microprocesses probably explains why it is difficult to use them to predict uprisings and why studies find contradictory results [79, 82].

In other words, structural models do not scale down well to the individual. Network models, on the other hand, show how small and large protests emerge from individual decision making. Since the same process can explain a greater range of outcomes, it should be preferred to ones that account for a smaller part of the protest distribution. Moreover, structural factors are usually used in conjunction with each other—a country with a geriatric leader, divided armed forces, and high inflation is more likely to have protests, for example—whereas a network model does not resort to several, possibly *ad hoc*, explanations. If individuals mobilize based on people they see mobilizing, the structure of connections between those individuals varies by country and time, and random small protests can occur, then large protests can also randomly occur.

### Discussion

Aggregating protest events with varied repression severity generates contradictory inferences because of idiosyncratic features of the events. Idiosyncratic features of protest events—who in a network has low participation thresholds, to whom they are connected, and the overall structure of the network, for example—means large protests are sometimes associated with severe repression and small protests with light repression, causing regressions to infer to that protest size increases as a function of repression. Because our simulation is set up so that repression should always decrease protest, the emergence of the repression-dissent puzzle suggests that it will remain a persistent feature of the empirical literature.

What, then, is to be done? The most obvious answer, and the empirical one pursued for decades, is to collect more data. Table 1, Figs 1 and 2, and the simulation results suggest,

however, that increasing amounts of data do not resolve the repression-dissent puzzle, even when simulations are structured so repression can only decrease protest size. While every dataset will put forth an internally consistent answer to the repression-dissent puzzle, that answer will likely not be consistent when bootstrapped within a dataset or analyzed across datasets.

Instead of collecting data on more protests, an alternative is to collect data about social networks before and during a protest. [14] uses ties within a social movement to estimate the probability that individuals cease participation; [12] shows protest information has differential effects depending on the degree centrality of its source, and [87] shows how to measure daily changes in online network structure; and [30] uses school attendance records to show how local network participation affects individuals' decision to protest. With more detailed data on a society's underlying network, the extent to which networks condition repression's effect may become clearer. New developments in survey methodology can elicit these data [34], and mobile phone data is another promising data source of social networks at scale [88].

A second alternative is to collect data about the distribution of participation thresholds in a population. We are not aware of any work that directly measures, even experimentally, protest participation thresholds, though existing studies may have the data to infer them [30, 89]. If these data are not available for a relevant population, then an estimate of the type of distribution could be used as city or country controls in a statistical model.

Finally, given the difficulty of collecting network and threshold data, efforts to understand repression and dissent could focus on making more narrow claims. It may be more productive to narrow repression's effect by regime type or technology available to protesters, for example. It could also be the case the repression's effect varies by year, just like the causes of political instability [79, 90] and armed conflict [91]. Instead of searching for a universal Truth, there may be many narrower Truths.

To borrow an argument about formal models and the study of war, the main theoretical task facing scholars of protest has been "not to add to the already long list of arguments and conjectures but instead to take apart and reassemble these diverse arguments into a coherent theory fit for guiding empirical research" [92, pg. 382]. This paper suggests that coherent theory may not be sufficient to untangle a puzzle at the heart of the repression-dissent literature. While clear theories, such as backlash [28] and networks as a conditioning variable [18], certainly explain some of the puzzle, the puzzle also results from measurement. Inconsistent results may be a feature, not a bug, of the repression-dissent literature.

## Supporting information

**S1 File.**
(PDF)

## Acknowledgments

We would like to thank the referees and anonymous reviewers for their constructive remarks. Megan Jones, William Karambelas, Aiqi Lin, and Aditya Voleti provided essential research support. At different points, Albert-László Barabási, Michael Chwe, Mirta Galesic, Jack Goldstone, Guy Grossman, Mark S. Handcock, Jennifer M. Larson, Howard Liu, Noa Pinter-Wollman, Mason A. Porter, David Siegel, T. Camber Warren, and Sebastian Ziaja provided valuable feedback. The same is true from participants at American Political Science Association, Human Nature Group, International Studies Association, Peace Science, Online Peace Science Colloquium, Political Networks, Santa Fe Institute, and European Political Science Association conferences.

## Author Contributions

**Conceptualization:** Shane Steinert-Threlkeld, Zachary Steinert-Threlkeld.

**Formal analysis:** Shane Steinert-Threlkeld, Zachary Steinert-Threlkeld.

**Investigation:** Shane Steinert-Threlkeld, Zachary Steinert-Threlkeld.

**Methodology:** Shane Steinert-Threlkeld, Zachary Steinert-Threlkeld.

**Resources:** Shane Steinert-Threlkeld, Zachary Steinert-Threlkeld.

**Software:** Shane Steinert-Threlkeld.

**Visualization:** Zachary Steinert-Threlkeld.

**Writing – original draft:** Zachary Steinert-Threlkeld.

**Writing – review & editing:** Shane Steinert-Threlkeld, Zachary Steinert-Threlkeld.

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
