## [Decision Letter · Decision Letter 0]

11 Feb 2021

PONE-D-20-37635

How Social Networks Affect the Repression-Dissent Puzzle

PLOS ONE

Dear Dr. Steinert-Threlkeld,

Thank you for submitting your manuscript to PLOS ONE. After careful consideration, we feel that it has merit but does not fully meet PLOS ONE’s publication criteria as it currently stands. Therefore, we invite you to submit a revised version of the manuscript that addresses the points raised during the review process.

We look forward to receiving your revised manuscript.

Kind regards,

Marton Karsai, PhD

Academic Editor

PLOS ONE

Additional Editor Comments (if provided):

As you see both expert reviewers found interesting your manuscript but both of the remarked that it is difficult to follow. In your resubmission, beyond addressing all raised points and comments by the reviewers, please pay special attention to make the structure and the text of the manuscript more accessible for the potential readers.

Journal Requirements:

2) We note that you have stated that you will provide repository information for your data at acceptance. Should your manuscript be accepted for publication, we will hold it until you provide the relevant accession numbers or DOIs necessary to access your data. If you wish to make changes to your Data Availability statement, please describe these changes in your cover letter and we will update your Data Availability statement to reflect the information you provide.

Reviewers' comments:

Reviewer's Responses to Questions

**Comments to the Author**

1. Is the manuscript technically sound, and do the data support the conclusions?

Reviewer #1: Yes

Reviewer #2: Partly

2. Has the statistical analysis been performed appropriately and rigorously? 

Reviewer #1: Yes

Reviewer #2: No

3. Have the authors made all data underlying the findings in their manuscript fully available?

Reviewer #1: Yes

Reviewer #2: Yes

4. Is the manuscript presented in an intelligible fashion and written in standard English?

Reviewer #1: Yes

Reviewer #2: Yes

5. Review Comments to the Author

Reviewer #1: The paper provides convincing evidence to the scientific debate on the repression-dissent puzzle. The authors are focusing on the effect of sampling, and ran a huge amount of different network simulations and models to show that inconsistent results are inherent features of the repression-dissent literature.

I found the paper particularly interesting and technically sound. Their major statement, that the repression-dissent puzzle cannot be solved at the level of general theories is supported by strong evidence, and moving the focus to making more narrow claims is an important conclusion of the paper. Based on this review, it has the potential to be accepted for publication in Plos One.

Before that, the authors are advised to consider the following comments and suggestions.

First of all, it is somewhat hard to follow the main line of thought when reading the paper. I am sure that future readers would reward some restructuring. The current structure is Introduction -> Results (Repression-Dissent Puzzle, The Model, Repression) -> Discussion. The text under "The Model" (that includes the majority of the text) might be separated with more sub-heading to help following the different applications of the model.

Also, I found the information in S5 (Implications for Understanding Repression and Protest) very important. Inclulding more direct references to the arguments in S5 in the main text could be useful.

A short explanation could added to the text describing the models where the network configuration parameters were adjusted to real social networks. Here my concern is how much the authors treat evidence from Twitter and email networks as proxies for real social networks? How were p and mt values adjusted to the clustering of the two networks?

Some adjustments in the size of the texts and some other minor changes in the figures should be made. (E.g. X axis text in Figure A3 and A4, more horizontal space between Fig6ab and Fig6cd; Fig5a and Fig5b ) In several figures the line type for insignificant models is literally indistinguishable from the positive ones.

Finally, I am not aware how this should be handled, but submission rules for Plos One state that "A Data Availability Statement describing where the data can be found is required at submission." Currently there is no reference in the text to the location of the data.

Reviewer #2: This paper's primary claim is that existing work on the effect of repression on dissent have been biased due to sampling approaches. To overcome this, the authors model protest and repression on social networks to show the effects from sampling subsets of the network space.

Overall, I think that this paper has an interesting idea and potentially compelling explanation at its core. But the paper is very hard to follow. For example, in the second paragraph on page two, in one paragraph, the authors say (1) the repression dissent puzzle might be driven by sampling issues; (2) so the authors will model protests on networks; (3) contextual factors like network structure and repression severity determine the success of protests, but inference is still sensitive to episodes. These are three very different sentences wedged into one paragraph. If the problem is about sampling, then is the solution about research design? Why networks?

The language here is so difficult to follow. By the end of page 2 the authors explain how their regressions will work, but it still isn't clear (a) why networks? when most of the lit has not studied networks-- this is a big leap to sell to your audience (repression/nonviolence scholars). Why is THIS decision important? And (b), what does it mean that the 'repression-dissent puzzle' here is 'fundamental' to the 'empirical approach' used in studies of protests and repression? The prose is hard to parse. Be clear, be straightforward, then we can interpret the meaning of your statistical analysis more clearly.

I liked the author's summary in Table 1, and discussion of the relevant literature but, again, there is very little attempt to explain to readers why the authors are going to take a 'network' approach. Though I did not re-read the studies cited, it seems they are predominantly event or country level data.

The discussion of the model should be moved to its own section. More discussion and depth is required as to why these three networks are chosen for the study. I know the authors moved some of this reasoning to the supplementary materials, but they could require these first 1-8 pages to be much clearer leading up to this point, especially since this seems to be essential to the main takeaway of the paper. Leading with networks, and their structure, early on will give the reader a better intuition about the paper.

The modeling section is clearer than other parts of the paper, but the empirical section can be revised to share more information about the data choices the authors made. There needs to be a clearer justification of the network parameters used in the model (eq 1) and the sensitivity of the results to these covariates. Further, it is unclear here which data the authors chose and why. Did they work with all the data listed in Table 1? Only a select amount? Again, the choices are buried.

It is unclear what the contribution is, following the simulations. If the problem of studying the repression-dissent puzzle is about measurement, then can the authors spell out exactly their contribution to solving this? On page 24 the authors suggest, the solution to the problem is to embrace a new research design, one that collect data on networks rather than events. This seems reasonable, and if it is the main argument then I encourage the authors to re-write it as the through-line of the paper. The authors can then use a burgeoning literature on intrastate conflict networks to support / compliment this idea (Larson, Dorff, Gade, Minhas, Cunningham etc).

6. PLOS authors have the option to publish the peer review history of their article (what does this mean?). If published, this will include your full peer review and any attached files.

Reviewer #1: No

Reviewer #2: No

---

## [Author Response · Author response to Decision Letter 0]

3 Mar 2021

Thank for the great reviewers and close reading. We have uploaded our responses as a response letter and do not paste them here because formatting would not be preservered.

---

## [Decision Letter · Decision Letter 1]

14 Apr 2021

How Social Networks Affect the Repression-Dissent Puzzle

PONE-D-20-37635R1

Dear Dr. Steinert-Threlkeld,

We’re pleased to inform you that your manuscript has been judged scientifically suitable for publication and will be formally accepted for publication once it meets all outstanding technical requirements.

Kind regards,

Marton Karsai, PhD

Academic Editor

PLOS ONE

Additional Editor Comments (optional):

Reviewers' comments:

Reviewer's Responses to Questions

**Comments to the Author**

1. If the authors have adequately addressed your comments raised in a previous round of review and you feel that this manuscript is now acceptable for publication, you may indicate that here to bypass the “Comments to the Author” section, enter your conflict of interest statement in the “Confidential to Editor” section, and submit your "Accept" recommendation.

Reviewer #1: All comments have been addressed

Reviewer #2: All comments have been addressed

2. Is the manuscript technically sound, and do the data support the conclusions?

Reviewer #1: Yes

Reviewer #2: Yes

3. Has the statistical analysis been performed appropriately and rigorously? 

Reviewer #1: Yes

Reviewer #2: Yes

4. Have the authors made all data underlying the findings in their manuscript fully available?

Reviewer #1: Yes

Reviewer #2: Yes

5. Is the manuscript presented in an intelligible fashion and written in standard English?

Reviewer #1: Yes

Reviewer #2: Yes

6. Review Comments to the Author

Reviewer #1: The authors adequately addressed all comments that I made in the first round of review. The structure of the paper has improved a lot, now it is easier to follow the main line of thoughts. Figures are also updated and the authors added a few external references to the issue I raised on using Twitter and email networks for modelling real social networks.

I also reviewed the replies and changes made for the request of R2 who asked for more clarifications and changes on the actual methodology. In my opinion the text also improved and became stronger in its argument and methodological foundations.

Reviewer #2: Excellent revision. I commend the authors for the great work they have completed on their MS--and during a pandemic, no less!

7. PLOS authors have the option to publish the peer review history of their article (what does this mean?). If published, this will include your full peer review and any attached files.

Reviewer #1: No

Reviewer #2: No

---

## [Editor Report · Acceptance letter]

22 Apr 2021

PONE-D-20-37635R1 

How Social Networks Affect the Repression-Dissent Puzzle  

Dear Dr. Steinert-Threlkeld:

I'm pleased to inform you that your manuscript has been deemed suitable for publication in PLOS ONE. Congratulations! Your manuscript is now with our production department. 

Kind regards, 

on behalf of

Dr. Marton Karsai 

Academic Editor

PLOS ONE